

# High-resolution maps of Arctic surface skin temperature and type retrieved from airborne thermal infrared imagery collected during the HALO–$(\mathcal{AC})^3$ campaign

Joshua J. Müller[1], Michael Schäfer[1], Sophie Rosenburg[1], André Ehrlich[1], and Manfred Wendisch[1]

[1]Leipzig Institute for Meteorology, University of Leipzig, Leipzig, Germany

**Correspondence:** Joshua J. Müller (joshua.mueller@uni-leipzig.de)

**Abstract.** Two retrieval methods for the determination of Arctic surface skin temperature and surface type based on radiance measurements from the thermal infrared (TIR) imager VELOX (Video airbornE Longwave Observations within siX channels) are introduced. VELOX captures TIR radiances in terms of brightness temperatures in the atmospheric window for wavelengths from 7.7 µm to 12 µm in six spectral channels. It was deployed on the High Altitude and LOng Range research aircraft (HALO)
during the HALO–$(\mathcal{AC})^3$ airborne field campaign conducted in the framework of the Arctic Amplification: Climate Relevant Atmospheric and SurfaCe Processes and Feedback Mechanisms $(\mathcal{AC})^3$ research program. The measurements were taken over the Fram Strait and the central Arctic in March and April 2022. To derive the surface skin temperature, radiative transfer simulations were performed assuming cloud-free atmospheric conditions, quantifying the influence of water vapour on the measured brightness temperature. Since this influence was negligible, it was possible to apply a single-channel retrieval of
the surface skin temperature. The derived surface skin temperatures were compared with data from the MODerate-resolution Imaging Spectroradiometer (MODIS). Furthermore, a pixel-by-pixel surface classification into types of open water, sea ice water mixture, thin sea ice, and snow-covered sea ice was developed using a random forest algorithm. When the resulting sea-ice concentrations are compared with satellite data, a mean absolute difference (MAD) of 5 % is obtained. In addition, the classified pixels were aggregated into segments of the same surface type, providing different segment size distributions for all
surface types. When grouped by the distance to the sea ice edge, the segment size distribution shows a shift, favoring fewer but larger floes in the direction of the pack ice.

## 1   Introduction

Arctic amplification comprises Arctic-specific processes and feedback mechanisms that cause the observed accelerated warming of the Arctic region as compared to the rest of the globe (Wendisch et al., 2023a). Another signature of Arctic amplification
involves the transition to fewer, thinner, and more dynamic sea ice within the last decades (Kwok, 2018; Meier and Stroeve, 2022; Budikova, 2009; Notz and Community, 2020). Therefore, observations of the current state of the Arctic sea ice are critical. Furthermore, the Arctic sea ice serves as a thermal insulator, regulating heat and moisture exchange between ocean and atmosphere (Maykut and Untersteiner, 1971; Qu et al., 2019). To quantify these exchange processes measurements of sea-ice surface skin temperature (IST) and open ocean sea surface temperature (SST) are crucial. In-situ measurements from buoys





or ship-borne instruments, are sparse in the Arctic due to harsh conditions and logistical challenges (Smith et al., 2019). As a
consequence, remote sensing techniques are used to determine IST and SST (Hall et al., 2004; Li et al., 2022; Nielsen-Englyst
et al., 2023). To retrieve these properties, established approaches use information supplied by observations in the wavelength
region of the atmospheric window ($7\,\mu m$ – $14\,\mu m$), where atmospheric absorption can mostly be neglected (McMillin and
Crosby, 1984; Liu et al., 2006). Specifically, wavelength bands centered around $11\,\mu m$ and $12\,\mu m$ are commonly used to re-
trieve the temperature of prevailing surface features (Hall et al., 2004). However, in the Arctic, these surface features partly
represent small-scale phenomena, such as leads, which are narrow openings in sea ice with spatial extents ranging from meters
to kilometers. Leads may account for a significant amount of net heat energy flux in the Arctic (Qu et al., 2019; Gryschka et al.,
2023). Additionally, melt ponds, which form on sea ice due to melting processes, reduce the surface albedo by up to $45\,\%$ (Tao
et al., 2024), thereby affecting the solar radiative energy budget (Anhaus et al., 2021; Niehaus et al., 2023). Established satellite
retrievals often lack the horizontal resolution needed to discriminate the majority of narrow leads and small melt-ponds. For
example, the horizontal resolution of the MODerate resolution Imaging Spectroradiometer (MODIS) onboard the Terra and
Aqua satellites (Willmes and Heinemann, 2015) restricts its observations to features larger than $500\,m$ (Hall et al., 2004). The
heterogeneous spatial distribution of typical Arctic surface types, e.g., open water, thin sea ice, snow-covered sea ice, melt
ponds, and transitional types plays an important role in the determination of the Arctic Radiative Energy Budget (REB) (Di Bi-
agio et al., 2021; Anhaus et al., 2021; Wendisch et al., 2023b).

To quantify spatial heterogeneity, surface classification algorithms have been developed, using empirically determined thresh-
olds (Massom and Comiso, 1994; Jäkel et al., 2019b; Thielke et al., 2023), including supervised (Wright and Polashenski,
2018) and unsupervised statistical learning approaches (Paul and Huntemann, 2021). Massom and Comiso (1994) used dif-
ferent wavelengths of thermal infrared (TIR) data from the Advanced Very High Resolution Radiometer (Cracknell, 1997,
AVHRR) to discriminate the surface into open water, new ice, young ice, and thick ice with a snow cover with a resolution
of $1.1\,km$ at nadir. The scene classification by Paul and Huntemann (2021) uses MODIS TIR data with wavelengths similar
to those in the approach of Massom and Comiso (1994), but the classification into open water, thin sea ice, thick sea ice, and
clouds is performed by a deep neural network instead of thresholds based on a histogram. As both approaches use TIR data,
they can be also applied at polar-night. In contrast, Jäkel et al. (2019b), Thielke et al. (2023),and Wright and Polashenski
(2018) rely on high-spatial resolution airborne data rather than satellite imagery. While Thielke et al. (2023) used a thermal
imager mounted to a helicopter during polar night, both Wright and Polashenski (2018) and Jäkel et al. (2019b) derived surface
type classifications with airborne based measurements in the visible wavelengths to distinguish open-water, melt-ponds, and
sea-ice. While Thielke et al. (2023) retrieved IST to distinguish sea-ice and open water with a resolution of $1\,m$, Wright and
Polashenski (2018) used imagery on the decimeter scale. In summary, satellite retrievals offer wide scene and consistent time
coverage, but are severely limited in their horizontal resolution, while airborne data mostly offers highly horizontally resolved
images, which are limited in their spatial and temporal coverage.

In this paper we describe the development of a skin temperature retrieval algorithm tailored for the TIR imager VELOX (Video
airbornE Longwave Observations within siX channels; Schäfer et al., 2022), combined with a surface type classification using
supervised machine learning techniques. A random forest algorithm (Breiman, 2001; Belgiu and Drăguţ, 2016; Wright and



Polashenski, 2018) was used to classify the observed surface types in a pixel-by-pixel fashion into four categories: Open-Water (OW), Ice-Water Mix (IWM), Thin Ice (TI), and Snow-Covered ice (SC). To further enable the interpretation of spatial properties of the surface types, a segmentation was applied unifying neighboring pixels of the same surface type into segments. The article is structured as follows: the measurements from the HALO–$(\mathcal{AC})^3$ campaign and satellite data used in this study are introduced in Sect. 2. The single-channel surface skin temperature retrieval method and the random forest algorithm used

to classify surface types are described in Sect. 3. The data are used to investigate the spatial characteristics of the classified surface types in Sect. 4.

## 2   Measurements and instrumentation

### 2.1   Airborne campaign

The HALO–$(\mathcal{AC})^3$ airborne campaign was conducted from 7 March to 12 April 2022, to investigate the evolution of air

mass transformation processes during warm air intrusions and cold air outbreaks in the Arctic by applying a unique quasi-Lagrangian approach (Wendisch et al., 2024). In total, 59 flights with multiple research aircraft were realized, among them 17 flights with research aircraft HALO, which was based in Kiruna, Sweden. In Fig. 1, the locations of measurement are depicted, together with the campaign-averaged sea-ice concentration (SIC). In addition to all HALO tracks that were flown during the campaign (limited to the map extent; for a full overview see Wendisch et al. (2024)). Due to HALO's range of up to

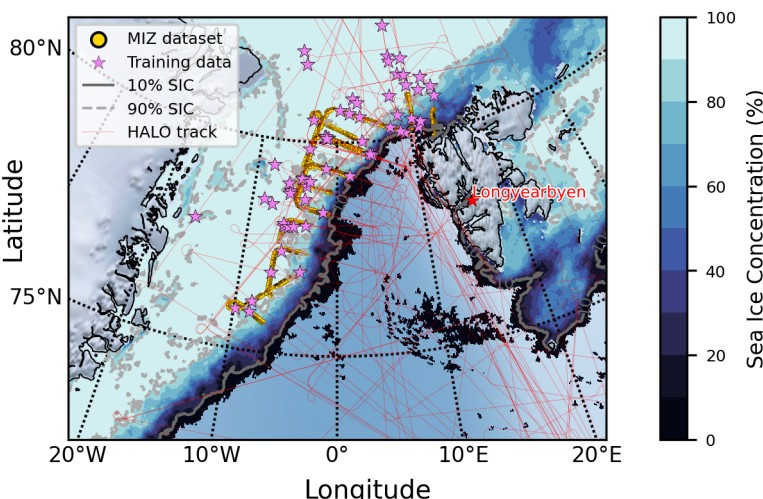

**Figure 1.** Overview of the data applied in this study, with the location of the data-points in orange and all flown HALO tracks in light red. The average SIC during the campaign is shown in blue contours, with a gray solid and dashed line indicating the 10 % and 90 % SIC contour, respectively. The data was provided by Spreen et al. (2008). Pink stars indicate the location of the training data used for the supervised classification.



9000 km, the measurements capture Arctic surface and atmospheric parameters on a regional scale while ensuring high spatial resolution when compared to satellite sensors like MODIS, AVHRR or the Sentinel-3 Sea and Land Surface Temperature Radiometer (SLSTR; Donlon et al., 2012) The variability of different Arctic surface types (open water, freshly formed sea-ice, snow-covered ice) is highest in the region between the ice-free open-ocean and the pack ice, resulting in a variability of the radiative energy budget (REB) due to their differences in radiative and thermodynamic properties. Here, we focus on

small-scale variability of the surface skin temperature resulting from the inhomogeneous distribution of Arctic surface types. Therefore, the following analysis will be restricted to the marginal sea ice zone (MIZ), which is suitable for such investigations. The MIZ is defined as the region where the campaign averaged sea-ice concentration (SIC) average was between 10 % and 90 %. In addition, we include data where the SIC exceeded 90 % for less than 10 min.

During the campaign, a set of remote sensing instruments was deployed on HALO (Ehrlich et al., 2024), of which only those

relevant to the analysis are briefly introduced here. As the main component of the instrumentation for this study, to capture two-dimensional (2D) fields of TIR radiance, the VELOX (Video airbornE Longwave Observations within siX channels) TIR imager was operated in a nadir-viewing configuration aboard HALO during the campaign. VELOX covers a spectral range of 7.7 to 12 μm, providing radiance measurements which are converted to brightness temperatures (Schäfer et al., 2022). At a typical flight altitude of 10 km, the imager achieves a horizontal resolution of 10 m by 10 m per pixel, corresponding to a

field of view (FOV) spanning an area of 5 km by 6 km. VELOX acquires images with a high temporal resolution of 100 Hz.
 The instrument is operated with six spectral filters resulting in six channels, of which two are redundant broadband channels

**Table 1.** Spectral wavelength range and thermal noise uncertainty in terms of the net equivalent temperature difference (NETD) of VELOX (Video airbornE Longwave Observations within siX channels) adapted from Schäfer et al. (2022)

| Channel | Wavelength range (μm) | Symbol | NETD (K) |
|---|---|---|---|
| 1 | 7.7–12.0 | $T_{B,1}$ | 0.048 |
| 2 | $8.7 \pm 0.6$ | $T_{B,2}$ | 0.347 |
| 3 | $10.7 \pm 0.4$ | $T_{B,3}$ | 0.605 |
| 4 | 7.7–12.0 | $T_{B,4}$ | 0.048 |
| 5 | $11.7 \pm 0.8$ | $T_{B,5}$ | 0.473 |
| 6 | $12.0 \pm 0.5$ | $T_{B,6}$ | 0.442 |

(channel 1 and 4). The remaining channels are narrow-band, each centered on specific wavelengths. The uncertainty in the measurements is characterized by the net equivalent temperature difference (NETD) for each channel. The broadband channels have a NETD of 0.048 K, while the narrow-band channels show varying NETD values, as shown in Table 1. For the airborne

field campaign HALO–$(\mathcal{AC})^3$, the corrected brightness temperature data, resampled to 1 s temporal resolution, was provided by Schäfer et al. (2023). To retrieve cloud cover, the HALO Microwave Package (Mech et al., 2014, HAMP) and the water vapour differential absorption lidar WALES (Wirth et al., 2009) were installed on HALO. In addition, more than 330 dropsondes (George et al., 2024) were released during the campaign. We have restricted our analysis to cloud-free scenes in the MIZ. For this purpose, a cloud mask based on campaign-specific radar reflectivity and lidar backscatter coefficient thresholds was





applied (Konow et al., 2019). To further ensure high data quality, each scene was visually examined to confirm the absence of clouds.

## 2.2 Satellite data

Independent measurements of surface skin temperature are provied by MODIS sea-ice surface temperature (IST; Hall and Riggs, 2021) and sea surface temperature (SST; NASA, 2024). Both data sets are based on a split-window retrieval algo-
rithm, which determined surface skin temperature from the measured brightness temperatures. For the respective surface types, MODIS channels 1 (0.645 μm), 2 (0.865 μm), 4 (0.555 μm), 6 (1.64 μm), 31 (11 μm), 32 (12 μm) were used. The datasets are provided with a horizontal resolution of 4 km by 4 km. Daily fields of SIC are provided by the assimilated MODIS/AMSR-2 SIC product, derived from a synthesis from MODIS and AMSR-2 (Ludwig et al., 2019). Depending on the combination of MODIS and AMSR-2, the fields of SIC have a 5 km horizontal resolution for all conditions, and 1 km for cloud-free scenes.
Satellite images in terms of spectral radiance with high horizontal resolution are obtained from the Sentinel-2 multispectral imager (MSI, hereafter Sentinel-2) data. To characterize the surface reflectivity, the red (0.664 μm), green (0.559 μm), and blue (0.492 μm) (RGB) channels are sufficient, which have a horizontal resolution of 10 m by 10 m. For the high latitudes reached by HALO-$(\mathcal{AC})^3$ observations, the revisit time of Sentinel-2 is about one day, which enables daily observations and allows for collocation of the satellite observations with VELOX images (Spoto et al., 2012). The Sentinel-2 data are accessed via the
Google Earth Engine (GEE; Gorelick et al., 2017).

## 3 Retrieval methods

### 3.1 Surface skin temperature

VELOX measurements provide TIR brightness temperature, $T_\mathrm{B}$ emitted by the surface (surface skin temperature). A significant contribution to the measured TIR $T_\mathrm{B}$ results from emission by atmospheric gases, although the spectral bands are located in the
atmospheric window region. Under Arctic conditions atmospheric emission results in a positive bias in the measured brightness temperature compared to the surface skin temperature. To correct for the atmospheric emission between the airplane and the surface, a split-window method (SW) is commonly applied (McMillin and Crosby, 1984; Li et al., 2013). Adjusted to VELOX measurements, this approach can be formulated as follows:

$$T_\mathrm{S} = a_\mathrm{sw} + b_\mathrm{sw} \cdot T_{\mathrm{B},5} + c_\mathrm{sw} \cdot (T_{\mathrm{B},5} - T_{\mathrm{B},6}). \tag{1}$$

Here, $T_\mathrm{S}$ represents the surface skin temperature, $T_{\mathrm{B},5}$ is the brightness temperature measured with VELOX channel 5 installed on HALO in about 10 km altitude, which is least affected by water vapour absorption. The coefficients $a_\mathrm{sw}$, $b_\mathrm{sw}$, and $c_\mathrm{sw}$ are empirically determined with a linear regression. Thus, $T_{\mathrm{B},5} - T_{\mathrm{B},6}$, represents the brightness temperatures difference between channels 5 and 6, serving as a proxy for water vapour absorption. Vincent et al. (2008) found that this brightness temperature difference observed in the Arctic region is also sensitive to other parameters, such as atmospheric inversion height or aerosol



particles. Their proposed single-channel algorithm (SCA; Vincent, 2019) can be adapted to VELOX as follows:

$$T_S = a_{sca} + b_{sca} \cdot T_{B,5}. \tag{2}$$

We have performed radiative transfer simulations (RTS) to constrain the contribution of atmospheric absorption to the surface skin temperature for both retrieval methods. The RTS were conducted with the radiative transfer library (*libRadtran*; Emde et al., 2016). The simulations were initialized with temperature and humidity profiles from dropsondes that were released from HALO during the campaign (Wendisch et al., 2024). The surface skin temperature is provided by MODIS (Hall and Riggs., 2021), whereas ozone content is given by the ERA5 reanalysis data (Hersbach et al., 2020). For the molecular absorption

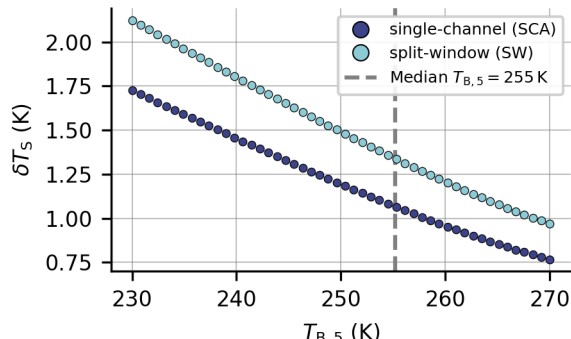

**Figure 2.** Empirically determined total uncertainty of the surface skin temperature retrieval $\delta T_S$ as a function of VELOX measured brightness temperature $T_{B,5}$. In dark blue, the total uncertainty was calculated for the SCA, in light blue for the SW.

parameters, *REPTRAN medium* (Gasteiger et al., 2014) was chosen, along with the DIScrete ORdinate Radiative Transfer solvers (*DISORT*; Stamnes et al., 2000). To constrain the retrieval uncertainties as a function of the atmospheric total column water vapour concentration, the integrated water vapour (IWV) was varied from $0\,\mathrm{kg\,m^{-2}}$ to $50\,\mathrm{kg\,m^{-2}}$. During the HALO–$(\mathcal{AC})^3$ campaign, the integrated water vapour (IWV) can be confined to values less than $10\,\mathrm{kg\,m^{-2}}$ (Walbröl et al., 2023). To evaluate the two retrieval methods the total uncertainty $\delta T_S$ is calculated for both algorithms. Adapted from Brown and Minnett (1999), who formulated the uncertainties for the MODIS IST retrieval, the total uncertainty of the VELOX retrievals can be formulated as follows:

$$\delta T_S^{sw} = \sqrt{(\delta T_{atm}^{sw})^2 + (\delta T_{VEL,i})^2}, \tag{3}$$

$$\delta T_S^{sca} = \sqrt{(\delta T_{atm}^{sca})^2 + (\delta T_{VEL,5})^2}, \tag{4}$$

$$\delta T_{vel,i} = \sqrt{(\delta T_{vel,i}^{sys})^2 + (\delta T_{VEL,i}^{ran})^2}, \tag{5}$$

where:

$$\delta T_{VEL,i}^{sys} = A + B \cdot T_{B,i}, \tag{6}$$

$$\delta T_{VEL,i}^{ran} = NETD_{B,i}. \tag{7}$$



The total uncertainty of the surface skin temperature $\delta T_{\mathrm{S}}$ is quantified as the square root of the sum of the squared uncertainties from the atmospheric correction, and the uncertainty introduced by VELOX $\delta T_{\mathrm{VEL},i}$, which can be split into a random part $\delta T_{\mathrm{VEL},i}^{\mathrm{ran}}$ equivalent to the NETD, and a systematic part $\delta T_{\mathrm{VEL},i}^{\mathrm{sys}}$. The systematic uncertainty was parametrized based on the measured brightness temperature $T_{\mathrm{B},i}$ in channel $i$. The superscripts 'sca' and 'sw' correspond to the respective algorithms, while the index subscript $i$ indicates VELOX channel $i$. The total uncertainty for both retrievals, depending on $T_{\mathrm{B},5}$ and

assuming a constant $\delta T_{\mathrm{atm}}$, is shown in Fig. 2. Due to difference in considering only the NETD of channel 5 for the SCA and both NETDs of channel 5 and 6 for the SW, the SCA retrieval has a lower error across all temperature ranges. To evaluate

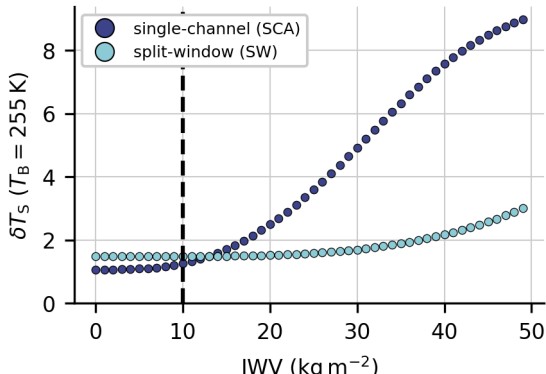

**Figure 3.** Comparison of surface temperature retrieval uncertainty ($\delta T_{\mathrm{S}}$) as a function of integrated water vapor (IWV, $\mathrm{kg\,m^{-2}}$) for single-channel algorithm (dark blue) and split-window method (light blue).

the sensitivity of the retrievals to IWV, the total uncertainty of both retrievals as a function of IWV is shown in Fig. 3. Below the IWV threshold of $10\,\mathrm{kg\,m^{-2}}$ the SCA outperforms the SW, due of the reduced NETD of using only one channel. Above this threshold, atmospheric absorption dominates the total uncertainty favoring the SW algorithm. In summary, the single-

channel algorithm has a lower total uncertainty for IWV values below $10\,\mathrm{kg\,m^{-2}}$, while the split-window algorithm is a more suitable for more humid atmospheres. Therefore, the SCA is applicable in the Arctic region when low IWV concentrations are present. As a consequence, we continue with the derivation of the regression coefficients $a_{\mathrm{sc}}, b_{\mathrm{sc}}$ of the SCA algorithm. For this purpose, the RTS were performed with a temporal resolution of one second, resulting in simulated brightness temperature values for VELOX channel 5 $T_{\mathrm{B},5,\mathrm{RTS}}$ at HALO flight altitude. These simulated brightness temperature values at flight altitude

are linearly regressed against MODIS surface skin temperature. The resulting fit parameters for slope and offset serve as the single-channel coefficients:

$$a_{\mathrm{sc}} = 9.051\,\mathrm{K} \tag{8}$$

$$b_{\mathrm{sc}} = 0.967\,\mathrm{K^{-1}}. \tag{9}$$

In Fig. 4, MODIS surface skin temperature $T_{\mathrm{S,MOD}}$ is plotted against the simulated brightness temperature $T_{\mathrm{B},5,\mathrm{RTS}}$. The

regression shows a coefficient of determination of $R^2 = 0.99$ and a root mean square error (RMSE) of 0.47 K. Substituting

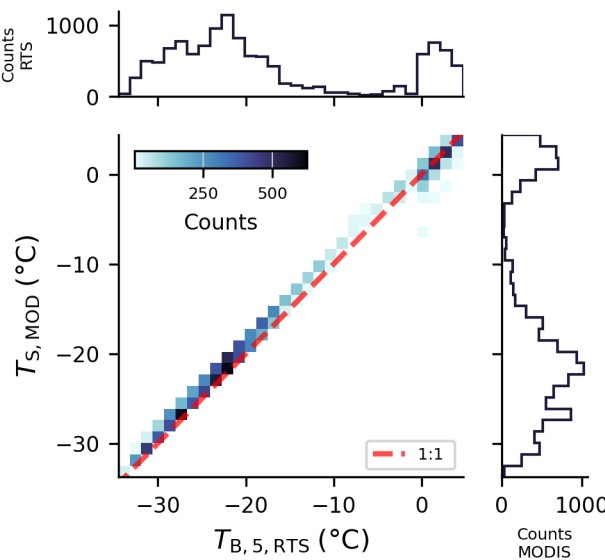

**Figure 4.** MODIS skin temperature $T_{\mathrm{S,MOD}}$ is compared to the simulated brightness temperature at HALO flight altitude for VELOX channel 5 (11.5 μm) $T_{\mathrm{RTS,B5}}$.

$\delta_{\mathrm{atm}}^{\mathrm{sc}} = 0.47\,\mathrm{K}$ and $\mathrm{NETD} = 0.473\,\mathrm{K}$ into Eq. 4, the total uncertainty of the SCA algorithm using RTS with ERA5 IWV data was computed to be:

$$\delta T_{\mathrm{S}} = 1.1 \pm 0.3\,\mathrm{K}, \tag{10}$$

where the range of $\delta T_{\mathrm{S}}$ reflects the error for different measurement conditions.

## 3.2 Surface type classification

To distinguish different surface types, we adapted established definitions (Miao et al., 2015; Wright and Polashenski, 2018; Jäkel et al., 2019a). As no melt-ponds were observed during HALO-$(\mathcal{AC})^3$, this surface type was omitted. To illustrate the surface types, a Sentinel-2 true color image is analyzed in Fig. 5. All surface types applied in this study are present in this scene and characterized in Tab. 2.

The image analysis consists of three steps. First, the VELOX 2D-images are preprocessed. Next, a random forest (RFA) classification algorithm is applied for pixel-wise surface typing. Finally, a segmentation algorithm is used to identify and summarized areas of same surface type.

### 3.2.1 Preprocessing images

Since the temporal sampling rate of the VELOX data applied here from Schäfer et al. (2023) is 1 Hz, and the typical cruise speed of HALO is about 200 m s$^{-1}$, it is possible to construct pushbroom-like images (PLI) of the corresponding nadir strips





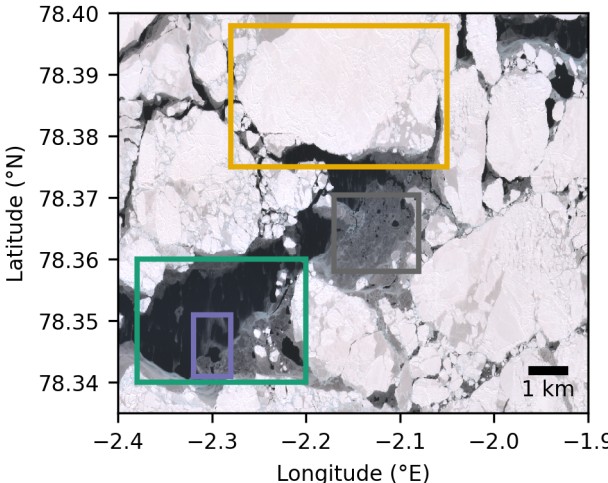

**Figure 5.** True color image provided by Sentinel-2 on 04.04.2022 at 13:35:00 local time. The four colored rectangles represent the surface types selected for this study: sea-ice free open-water (green), ice-water mix (purple), thin sea ice (gray), and snow-covered sea ice (yellow)

at each timestep. With this technique, the effect of the viewing azimuth angle (VZA) on the measured brightness temperature can be neglected. Furthermore, georeferencing is performed for each data point of the PLI, providing crucial information about the geographic location of the measurement. This process incorporates the geographical position, flight altitude, and attitude data from HALO, as well as the calculated viewing azimuth and zenith angles for the applied lens and detector combination of VELOX, and the measured mounting direction of VELOX.

### 3.2.2  Random forest classification

To determine the surface type, a random forest (RFA) was implemented in a pixel-by-pixel fashion, i.e., each pixel is analyzed individually. The RFA is a supervised machine learning method that constructs ensembles of decision trees, which are fitted to user-defined ground-truth data. It combines the interpretability of decision trees with the robustness to noise characteristics of other ensemble methods (Breiman, 2017; James et al., 2023). For the implementation of the RFA, the machine learning library *autogluon* (Erickson et al., 2020) was used, allowing for a comparison of multiple machine learning methods. Compared to other supervised learning algorithms, the RFA demonstrated comparable accuracy, while significantly reducing computation time, suggesting it as a preferable choice (not shown).

Sentinel-2 images classified manually were used as the ground truth. For labeling these images, the Computer Vision Annotation Tool (Sekachev et al., 2020, CVAT) was used. In total, 58 VELOX images from 10 different research flights were labeled, resulting in over 13 million labeled pixels. The locations of the training data are depicted as pink stars in Fig. 1. The training data were sampled randomly from the available data (Sentinel-2 image available, cloud free) and subsequently filtered to resemble all latitudes equally. Seven input features, as defined in Tab. 3, are applied to the RFA. All parameters are



**Table 2.** Surface types with short characterization, corresponding abbreviations, and representative images from Fig. 5.

| Abbr. | Surface Type | Image |
|---|---|---|
| **OW** | Open Water: sea-ice free surfaces of open ocean water, including leads. | |
| **IWM** | Sea Ice-Water Mixture: unconsolidated frazil and grease ice, mixed with open ocean water. | |
| **TI** | Thin Sea Ice: freshly formed sea ice (nilas), appearing dark or grey in optical wavelengths. | |
| **SC** | Snow-Covered Sea Ice: sea-ice covered with a snow layer. | |

calculated from VELOX brightness temperature data. The accuracy of a multi-class classification problem can be expressed by
the ratio of correct to all predictions. When validated in five-fold cross-validation setup, the RF showed an accuracy of 87 %
with respect to the test data. To further assess the performance of the RFA, a confusion matrix is shown in Fig. 6. The highest
accuracy is achieved on the SC surface type (95 %), followed by the OW surface type (90 %). The TI surface type achieves a
lower overall accuracy with 71 %, due to transitional nature of this surface type. In Fig. 7 b) the initial RFA classification for an
example scene is shown. A common challenge when using RFA for image classification are the speckles, as seen in this figure.
To address this issue, segmentation is required, which is described in the next section.





**Table 3.** Input parameters as processed from VELOX measurements and used in the pixel-wise RF surface type classification.

| Variable | Description |
| --- | --- |
| $T_{B,1}$ | VELOX channel 1 (7.7 µm to 12 µm). |
| $\Delta T_{B,2\text{-}5}$ | Brightness temperature difference (BTD) between channels centered at 8.54 µm and 11.7 µm. |
| $\Delta T_{B,3\text{-}5}$ | BTD between channels centered at 10.7 µm and 11.7 µm. |
| $\Delta T_{B,5\text{-}6}$ | BTD between channels centered at 11.7 µm and 12 µm. |
| $\lvert \nabla T_{B,1} \rvert$ | Magnitude of the horizontal gradient of broadband brightness temperature as a measure of horizontal inhomogeneity. |
| $\overline{T_{B,1}}$ | Mean of $T_{B,1}$ in a $5 \times 5$ pixel neighborhood. |
| $\sigma T_{B,1}$ | Standard deviation of $T_{B,1}$ in a $5 \times 5$ pixel neighborhood. |

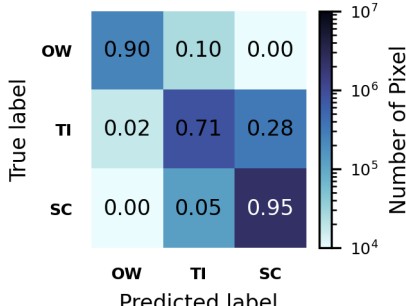

**Figure 6.** Confusion matrix of the RFA prediction, showing the percentage of the correctly predicted pixels on the diagonal. The off-diagonal elements represent the false positive and false negative values.

### 3.2.3 Segmentation

To assign the predefined surface types from Sect. 3.2 to the retrieved fields of surface skin temperature, the PLIs are subjected to the open-source image segmentation algorithm *segment-anything* (Kirillov et al., 2023, SAM;). The SAM algorithm image segments on the basis of color-gradients and points that are placed by the user. The initiall segmentation of an exemplary scene is shown in Fig 7 c). Although the model was not fine-tuned, i.e., not trained with a specific user dataset, it proves a high capability to segment previously unseen data in a zero-shot fashion (Wu and Osco, 2023; Ren et al., 2024). This offers an advantage over training for a segmentation algorithm, which is often demanding in terms of data points and computational




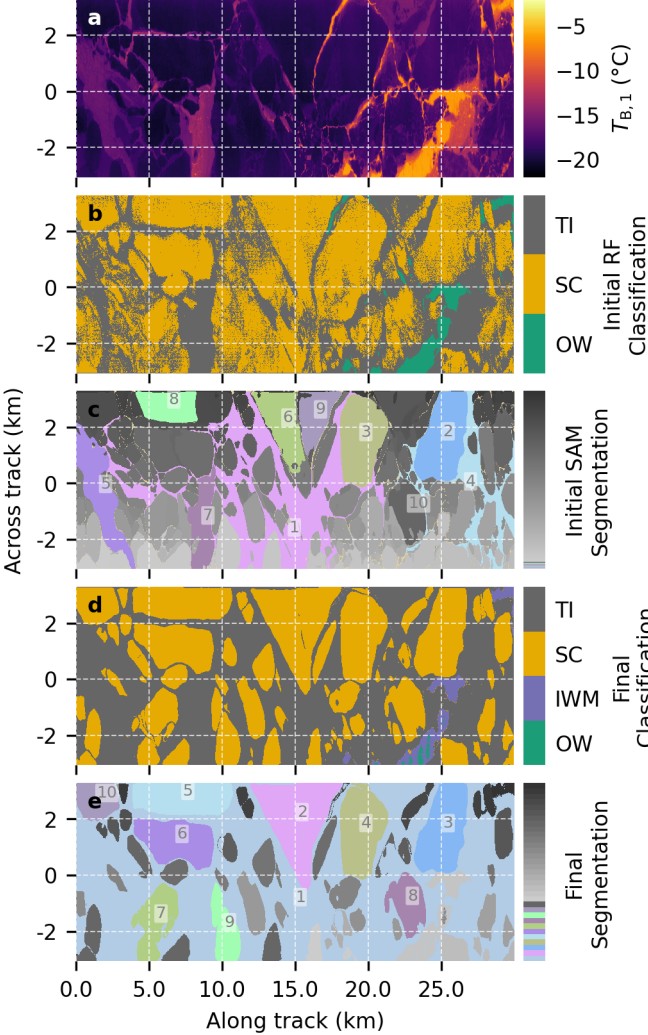

**Figure 7.** Overview of surface classification and segmentation results for the pushbroom-like image captured on 04. April 2022 from 13:36:14 to 13:38:31 UTC. a) Broadband brightness temperature $T_{B,1}$ (7.7 µm–12 µm) as pushbroom-like image b) Initial surface type classification using the random forest algorithm (RFA), identifying open-water, thin ice, and snow-covered ice. c) Initial segmentation using the *segment-anything* model (SAM), with numbered segments representing the ten largest areas for illustration. d) Final surface type classification: the most common surface type within each segment from (c) was assigned, and a surface skin temperature threshold was used to sort the ice-water mix class (IWM) form OW. (e) Final segmentation, where new segments were assigned to all connected regions of the same surface type derived from (d), with the largest segments again highlighted by their respective numbers.




time.

To automatically generate a segmentation mask with SAM, a grid of points is placed on the PLI and then recursively shifted
to avoid over-segmentation of the images. This means, that initially a grid is constructed on the image, and the algorithm
searches for segments close to the gridpoints. To ensure stable segmentation, the grid is divided into smaller subgrids, which
are then shifted relative to the initial grid points. This process is repeated three times. Since some over-segmentation still
occurs, resulting in smaller predicted segments than those identified by humans, information from surface classification is
added to the segmentation. First, each segment identified by SAM is subjected to a majority vote, meaning the most frequently
occurring surface type within a particular segment is assigned to that segment. Finally, the segments are obtained by merging
neighboring segments of the same surface class. This results in a natural image segmentation, which is illustrated in the lowest
panel of Fig. 7 e). To further ensure the quality of Open-Water classified pixels, a threshold is applied sorting all pixels cooler
than -2.5 °C into the Ice-Water Mix Class

## 4 Results and Discussion

### 4.1 Surface skin temperature analysis

In Fig. 8, the VELOX retrieved surface skin temperature $T_{S,VELOX}$ is compared to the MODIS surface skin temperature $T_{S,MOD}$,
obtained from Hall and Riggs. (2021) and NASA (2024), showing the coefficient of determination $R^2$ to be equal to 0.96.

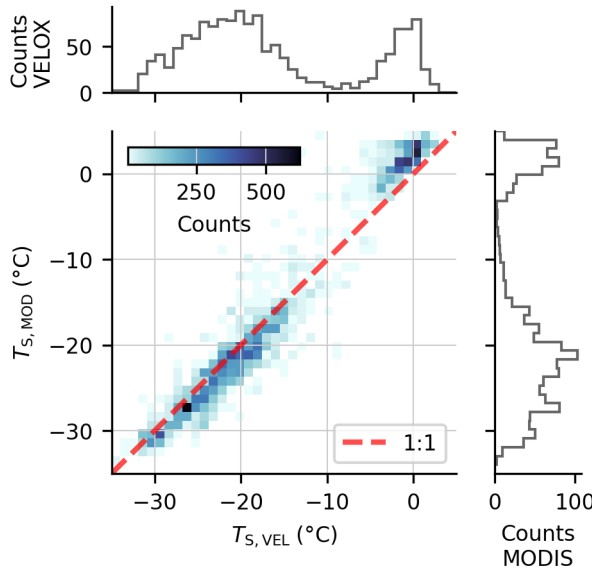

**Figure 8.** Scatter plot of MODIS $T_{S,MOD}$ against VELOX retrieved $T_{S,VEL}$, with VELOX data averaged to match the MODIS pixel size.
The appended frequency distributions show the corresponding surface skin temperature distributions for both datasets.




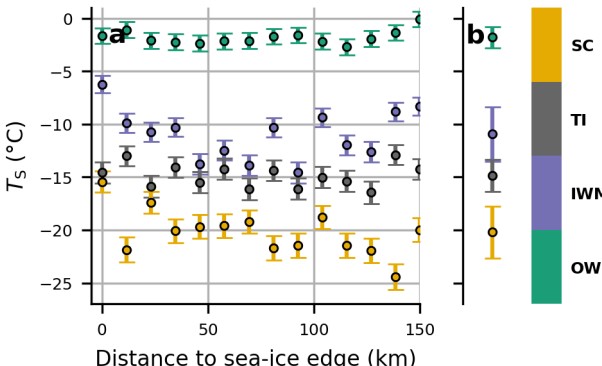

**Figure 9.** a) Mean $T_S$ of different surface type segments, weighted by segment size and aggregated over 10 km bins from the sea-ice edge into the internal ice zone. b) Mean $T_S$ over all Marginal sea-ice zone segments.

For this comparison, the instantaneous FOVs of the single VELOX pixels were combined by averaging to fit the MODIS pixel-size, allowing for a direct comparison between their two datasets. The RMSE was determined to be 2.4 K with a bias

of -2.4 K, indicating a underestimation of surface skin temperature by VELOX, $T_{S,VEL}$, with respect to MODIS, $T_{S,MOD}$. As the data set comprises multiple days, it is essential to provide information on the location of the data. To simplify this spatial information into a scalar, the data are grouped by its distance to the sea-ice edge (positive direction into the internal ice-zone). As the individual pixels have been georeferenced, their relative distance to the nearest sea ice edge (defined by campaign-averaged SIC values between 9-11 %) is computed. For this, the distance of each spatial segment center to the temporally

closest available AMSR-2/MODIS SIC pixel is calculated. The mean surface skin temperature colored by surface types is plotted in Fig. 9a against the distance to the sea-ice edge. In Fig. 9b, the mean surface skin temperature of all segments $\overline{T_S}$, weighted by their size is shown. A clear separation between the $\overline{T_S}$ of different surface types is observed as expected. The values from Fig. 9b are displayed together with the corresponding error range in Tab. 4.

## 4.2  Spatial analysis of surface types

From the segmentation, we retrieve the corresponding segment size, mean temperature and standard deviation of each segment. The results are illustrated in Fig. 10, showing the spatial distance of each segment center to the nearest sea-ice edge plotted against the corresponding surface type. The fraction of the open water surface type-fraction decreases from 40 % to below 5 % in the first 20 km, while the fraction of the snow-covered surface type-fraction becomes increasingly dominant when approaching the pack-ice. The thin-ice and ice-water-mix surface types maintain relatively constant fractions of occurrence

across all considered distances from the ice edge, with no clear trend observed. When compared with the provided SIC from MODIS/AMSR-2, the computed RMSE and mean absolute difference (MAD) are 8 % and 5 %, respectively. For this comparison, the nearest available SIC data from the satellite product were matched with a similar FOV of the VELOX PLI. The errors result from the temporal mismatch between both data sets, as MODIS/AMSR-2 SIC is only available as daily gridded product.



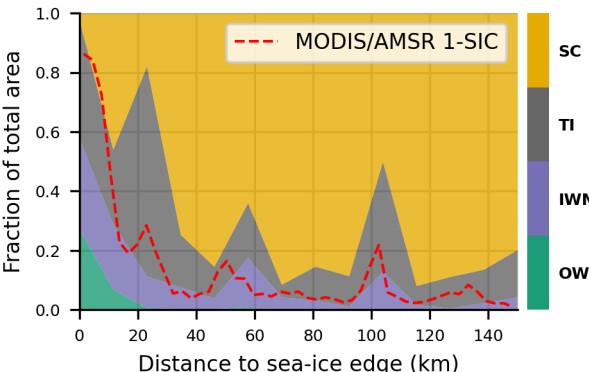

**Figure 10.** Fraction of total area for the four surface classes as a function of the distance to the closest sea-ice edge. The red dashed line indicates the open water fraction (1.0 - SIC) from MODIS/AMSR-2 (Ludwig et al., 2019).

When comparing the bias, i.e., the difference between VELOX SIC and MODIS/AMSR-2 SIC, an underestimation of 3 % or an overestimation of 5 % is observed, depending on whether only open-water classified pixels are considered as open water or if both open-water and ice-water-mix classified pixels are included. As shown in Fig. 10, the open-water fraction and the area fractions derived from VELOX agree within the given error range.

## 4.3 Segment size distribution

Since the segmentation enables the measurement of individual segment sizes, an analysis of the spatial structure of the data is performed. Here, we extend the concept of the floe size distribution (FSD; Rothrock and Thorndike, 1984; Herman, 2010; Bateson et al., 2022) to the segment size distribution $N_{\mathrm{SSD}}$, resulting in the following description:

$$N_{\mathrm{SSD}}(x_{\mathrm{SEG}}) = C \cdot (x_{\mathrm{SEG}})^{\beta}. \tag{11}$$

Here, $x_{\mathrm{SEG}}$ represents the segment size in units of m$^2$, $C$ is an empirical constant, and $\beta$ is the dimensionless power-law exponent describing the scaling of the distribution. The closer the exponent is to zero, the more $N_{\mathrm{SSD}}$ favors large segments. This approach simplifies the complex spatial heterogeneity of the MIZ by expressing the scaling of $N_{\mathrm{SSD}}$ using $\beta$, a single scalar value In Fig. 11, the segment size density $N_{\mathrm{SSD}}$ (in units of km$^{-2}$) for different surface types is displayed in a double-logarithmic graph as a function of the segment size size, $x_{\mathrm{SEG}}$. In addition, the individual distributions are fitted with a linear model. The slope of each linear fit corresponds to the exponent of the power-law distribution, $\beta$. The different $\beta_i$ computed for each surface type are shown in Tab 4. We conclude that, in addition to the different sea-ice types, the ice-water-mix and open-water surface types also follow a power-law distribution. The computed $\beta$ for, e.g. the snow-covered sea-ice type are in the range of corresponding literature data, with values ranging from -0.91 to -2.9 (Herman, 2010). To gain more insight into the spatial heterogeneity within the MIZ, we fit the $N_{\mathrm{SSD}}$ of the snow-covered segments to 10 km sized bins of distance to the sea-ice edge (in pack-ice direction). In Fig. 12, the size power-law exponent $\beta$ is shown as a function of the distance to the



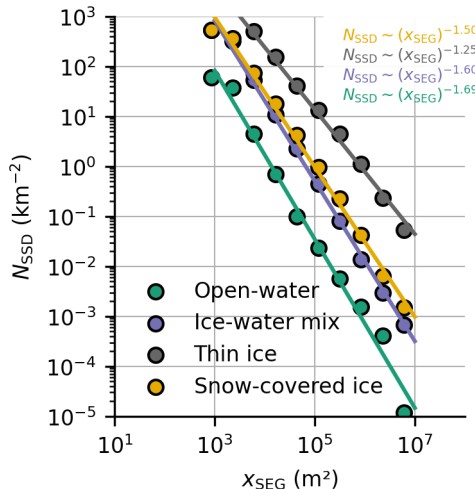

**Figure 11.** Double-logarithmic graph of the segment size density (colored dots) for all four surface types as a function of the segment size. The surface types are color-coded, indicating open water (green), ice-water-mix (purple), thin ice (grey), and snow-covered ice (yellow). Linear fits (colored lines) are added in the respective surface type color, providing the exponents listed at the top right.

**Table 4.** Summary of mean surface temperature $\overline{T_\mathrm{S}}$, power-law exponents $\beta$, and goodness of fit $R^2$ for the corresponding $N_\mathrm{SSD}$, for different surface types.

|  | $\overline{T_\mathrm{S}}$ | $\beta$ | $R^2$ |
|---|---|---|---|
| **OW** | $-3.2 \pm 1.1$ | $-1.68 \pm 0.04$ | 0.987 |
| **IWM** | $-12.1 \pm 2.3$ | $-1.60 \pm 0.02$ | 0.992 |
| **TI** | $-17.0 \pm 1.4$ | $-1.25 \pm 0.02$ | 0.996 |
| **SC** | $-22.2 \pm 2.0$ | $-1.50 \pm 0.02$ | 0.992 |

sea-ice edge for the different surface types. A linear trend is fitted only to the TI data, suggesting significance with a $R^2 = 0.76$
275 and a p-value less than 0.001. The increase in $\beta$ from -1.6 to -1.3 reflects a physical characteristic of the MIZ. Closer to the sea-ice edge, a higher number of smaller segments is observed (more negative $\beta$) due to intensified floe breakup, whereas larger floes (less negative $\beta$) become more prevalent further into the MIZ, where ocean wave propagation is more attenuated (Herman, 2010; Denton and Timmermans, 2022).

## 5 Summary and conclusions

280 During the HALO–$(\mathcal{AC})^3$ airborne field campaign, conducted in the area from the Fram Strait to the North Pole in March and April 2022, an extensive dataset of surface and atmospheric properties was collected from a variety of instruments mounted on three research aircraft (Wendisch et al., 2024). Here we use data compiled by the High Altitude and LOng range research

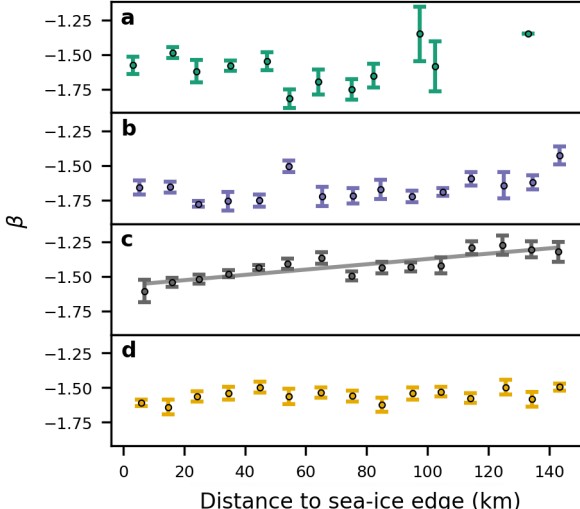

**Figure 12.** Power-law exponent $\beta$ of the segment size distribution $N_{\mathrm{SSD}}$, binned in $10\,\mathrm{km}$ steps, starting from the sea-ice edge into the direction of the internal ice zone by surface type. For panel a) open-water, b) ice-water mix, and d) snow-covered ice, no trend is observed. For panel c) thin ice, a significant linear trend is observed.

aircraft (HALO), which was instrumented with radar, lidar, a dropsonde launching facility, microwave radiometer, and various spectral imagers. Here we have used data collected by the VELOX (Video airbornE Longwave Observations within siX channels; Schäfer et al., 2022) thermal infrared (TIR) imaging system, which was installed On HALO in a nadir viewing direction. Due to its fast-spinning filter-wheel ($100\,\mathrm{Hz}$) equipped with multiple spectral band-pass and long-pass filters, a high spatial resolution of $10\,\mathrm{m}$ by $10\,\mathrm{m}$ pixel size for a target at $10\,\mathrm{km}$ distance is achieved with VELOX providing valuable high-resolution TIR spectral radiances expressed in brightness temperatures.

Using VELOX data from HALO–$(\mathcal{AC})^3$, which are publicly available from Schäfer et al. (2023), a single-channel (SCA) surface skin temperature retrieval based on linear coefficients derived from radiative transfer simulations (RTS) is adapted. Comparisons with multiple-channel retrievals and surface skin temperature products from the MODerate resolution Imaging Spectroradiometer (MODIS; Hall et al., 2004; Hall and Riggs., 2021; NASA, 2024), provide convincing agreement, with a coefficient of determination of $R^2 = 0.96$ and a bias of -2.4 K. To categorize the obtained surface skin temperature fields, a surface type classification algorithm is developed based on publicly available software tools combined with physically reasonable thresholds applied to regenerated push-broom images from the initial brightness temperature data. The resulting two-dimensional fields provide segment-vise information of the surface type, which can then be analyzed in combination with , e.g., the retrieved surface skin temperature. The data are classified into Open-Water (OW), Thin-Ice (TI), Ice-Water-Mix (IWM), and Snow-Covered-Ice (SC) surface type.

With the surface skin temperature and surface classification retrievals, important parameters with high spatial resolution are obtained. When computing the resulting Sea-Ice Concentration (SIC) from the surface classification, a reasonable agreement

(bias $\approx 5\,\%$) with the MODIS/AMSR-2 product is achieved, if the IWM surface type is assigned to be "sea-ice-free". Additional sensitivity studies will be required to assess the influence of this surface type. The established classification serves as a promising foundation for these investigations. The retrieved power-law segment size statistics are generally consistent with values reported in the literature (Denton and Timmermans, 2022). For the snow-covered surface type, these findings align

with those of Herman (2010), who observed power-law exponents ranging from -0.91 to -2.9 and reported an increase in the exponent when transitioning from the sea-ice edge to the interior ice zone. Overall, although the temporal duration and spatial extent of the presented data set is limited, the agreement with other studies emphasizes its value for the sea-ice community. Future analysis will focus on leads in the inner Arctic, using the presented methods.

*Data availability.* The 2D VELOX brightness temperature data can be accessed at https://doi.org/10.1594/PANGAEA.963401. The surface

skin temperature, surface type and segment data are in submission to pangaea. We thank the Institute of Environmental Physics, University of Bremen for the provision of the merged MODIS-AMSR2 sea-ice concentration data at https://data.seaice.uni-bremen.de/modis_amsr2 (17.12.2024).

*Author contributions.* JM, MS, SR, AE, and MW contributed to the conception and design of the study. JM elaborated the methods, performed the analyses, created the figures, and prepared the original draft. All authors discussed the results, contributed to manuscript revision

and approved the final submitted version.

*Competing interests.* Some authors are members of the editorial board of AMT.

*Acknowledgements.* We gratefully acknowledge the funding by the Deutsche Forschungsgemeinschaft (DFG, German Research Foundation) – Project Number 268020496 – TRR 172, within the framework of the Transregional Collaborative Research Center "ArctiC Amplification: Climate Relevant Atmospheric and SurfaCe Processes, and Feedback Mechanisms $(\mathcal{AC})^3$". The authors are thankful to AWI for providing

and operating the two Polar 5 and Polar 6 aircraft. We thank the crews and the technicians of the three research aircraft for excellent technical and logistical support. The generous funding of the flight hours for the Polar 5 and Polar 6 aircraft by AWI, and for HALO by DFG, Max-Planck-Institut für Meteorologie (MPI-M), and Deutsches Zentrum für Luft- und Raumfahrt (DLR) is greatly appreciated. We are further grateful for funding of project grant number 316646266 by DFG within the framework of Priority Program SPP 1294 to promote research with HALO. This publication was supported by the Open Access Publishing Fund of Leipzig University.



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
