# Peer review of "High-resolution maps of Arctic surface skin temperature and type retrieved from airborne thermal infrared imagery collected during the HALO– $(\mathcal{AC})^3$ campaign"

_Atmospheric Measurement Techniques, 2024_

## Author Response (AR1)

The authors thank the reviewer for their insightful comments and suggestions that have helped us to improve the manuscript. We have carefully considered all the points raised and have revised the manuscript accordingly. In the following, we provide a point-by-point response to the reviewers' comments. Further, we proof-read the manuscript again and made minor changes for readability.

Original comments from the reviewers are shown in gray.

**Authors replies are shown in bold.**

*Excerpts from the manuscript are show in italic.*

**CC1:**

The manuscript present analysis Arctic surface skin temperature and surface type, obtained from airborne thermal measurements during HALO-(AC)3. In general, the manuscript provide updated assessment of sea ice surface in the marginal ice zone (MIZ) using high-resolution airborne thermal images, which will promote research on sea ice dynamic and thermodynamic process in the MIZ.

Although, i have a few minor question for the author

1. It seems you used Level 3 MODIS daily IST for radiative transfer simulations and evaluation of airborne IST algorithm, and you mentioned the temperal mismatch between airborne and satellite data. Assuming there were 1 or 2 HALO flight in one day during the compaign, have you tried Level2 MODIS swath data, i.e. MOD29/MYD29 instead of Level 3 gridded product? the swath data suffer more from cloud contamination of course, but it  will help reduce the time different to less than an hour, and possibly, you would find better agreement between airborne and satellite data.

**We thank the reviewer for this suggestion. We agree that using MODIS Level 2 swath data (MOD29/MYD29) is a more suitable approach to minimize the temporal mismatch between the airborne and satellite observations. Following the reviewer's advice, we have re-analyzed our data using available cloud-free MODIS Level 2 scenes that coincided with our HALO flight tracks. This new comparison has**

**yielded an improvement in the agreement between the datasets. While the correlation coefficient remained the same, the root mean squared error (RMSE) was reduced from 2.49 K to 2.00 K, and the mean absolute error (MAE) decreased from 1.92 K to 1.55 K. Fig. S1 shows the updated comparison between MODIS Level 2 swath data and VELOX data. We have updated Section 2.2, Figure 8, and the**

**30    corresponding discussion in Section 4.1 of the manuscript to reflect these improved results.**

[Figure]

Figure S1: MODIS Surface skin temperature from the MOD29 Product with 1km horizontal resolution compared to co-located VELOX surface skin temperature. This figure will replace Fig. 8 in the original manuscript.

*Ln 106: The IST dataset is provided as swaths with horizontal resolution of 1 km by 1 km, while the SST dataset is gridded with a horizontal resolution of 4 km by 4 km.*

*Ln 232: The RMSE was determined to be 2.0 K with a bias of 0.51 K and a mean average*
*difference of 1.55 K. Further, Fig. 8 indicating slightly higher values of surface skin temperature by VELOX, $T_{S,VEL}$, with respect to MODIS, $T_{S,MOD}$ over sea-ice and lower vales over open-water.*

*Ln 303: Comparisons with multiple-channel retrievals and surface skin temperature products from the MODerate resolution Imaging Spectroradiometer (MODIS; Hall et al.,*
*2004; Hall and Riggs., 2021; NASA, 2024), provide convincing agreement, with a coefficient of determination of $R^2$ = 0.96 and a bias of 0.5 K.*

2. if i get it right, the class Sea Ice-Water Mixture (IWM) is not included in the training set, but only classified Open-water pixels with surface temperature below -2.5°C. it seems the IWM only account for a minor portion of area (Fig.7), and its surface temperature is
very close to thin ice(TI). the question is, how is the threshold determined? Also, since you used Sentinel-2 MSI for labeling of the images in training set, It is recommended to check the distribution of corresponding reflectance (from MSI) for different surface class in the training set, IWM samples might appear in the OW samples as anomaly.

**We thank the reviewer for this question regarding the classification of the Ice-Water**
**Mix (IWM) class. The reviewer is correct that the IWM class was not part of the**
**initial training set but was identified in a post-processing step using a temperature**
**threshold. This threshold of −2.5°C was established to correct instances where the**
**Random Forest Algorithm classified pixels as Open Water (OW) despite having**
**temperatures significantly below the physical freezing point of seawater. Given that**
**the freezing point is approximately −1.7°C (Skogseth et al., 2009; De La Rosa et al.,**
**2011) and our measurement uncertainty is about 0.8 K in this temperature range,**
**we chose the −2.5°C threshold to reliably reclassify these pixels as IWM while**
**accounting for measurement variability. Although the case study in Fig. 7 of the**
**manuscript implies some temperature overlap, the dataset-wide statistics confirm**
**a clear distinction between the classes. The average surface skin temperature for**
**IWM segments is −12.1 ± 2.3°C, while for OW segments it is −3.2 ± 1.1°C. To further**
**visualize the overall surface skin temperature distribution of the full dataset and**
**the temperature threshold, we refer to Fig. S2. Following the suggestion of the**

[Figure]

Figure S2: Surface skin temperature distribution for all data points.

[Figure]

Figure S3: Threshold for IWM classification as function of sub-pixel mixtures of different surface types,
assuming a constant OW surface skin temperature and varying sea-ice sub-pixel skin temperature.

**reviewer, we have examined the Sentinel-2 MSI reflectance characteristics for the**

**surface types used in our training. We used Sentinel-2 imagery as a visual reference to label our training data, as illustrated in the example scene in Figure 5 of the manuscript. In response to the comment, we have provided the Sentinel-2 MSI reflectance distributions for our example scenes in the supplementary material (Fig. S3).**

**This analysis shows that, while the reflectance patterns for IWM and OW are similar, there is a discernible difference, particularly in MSI Band 2 (blue), although these differences are small. This finding reinforces the utility of the IWM class. It helps resolve the ambiguity where a surface may appear dark (similar to open**

**water) but has a skin temperature well below freezing, indicating the presence of newly formed, unconsolidated ice. We believe that this post-processing step improves the physical consistency of our final classification product.**

[Figure]

Figure S3: Distributions of normalized reflectance from Sentinel-2 MSI for the Scenes shown in Table 2 of the original manuscript.

*Ln 200: In a final post-processing step, the Ice-Water Mix (IWM) class is identified to ensure the physical consistency of the final product. This step addresses instances*

*where pixels are classified as Open Water (OW) despite having temperatures well below the physical freezing point of seawater. Specifically, any OW pixel with a surface skin temperature cooler than -2.5 °C is reclassified as IWM. This threshold was chosen to be significantly lower than the approximate -1.7 °C freezing point of seawater (Skogseth et al., 2009; De La Rosa et al., 2011), thereby accounting for the total uncertainty of our*

*surface skin temperature retrieval, which was computed to be less than 0.8 K for temperatures around the freezing point.*

3. about the size distribution of surface features in the MIZ, it seems the thin ice area is connected into a very large segment in Fig.7. i'm not sure if is it a common phenomenon in your data set, but it seems the TI is the dominant class in Fig.11. It is recommended to break down the large TI segment into pieces by any means, otherwise the FSD/SSD result for TI could be misleading.

**We appreciate the reviewer's observation regarding the large, connected segments of thin ice (TI) and the potential for this to influence the feature size distribution (FSD) analysis. The presence of large, contiguous areas of newly forming ice was a characteristic feature observed during our flights over the marginal ice zone (MIZ). We interpret these large segments as genuine features of the MIZ physics at the time of observation, rather than segmentation artifacts. We agree that the definition of a "segment" can be challenging, and any method to artificially break up these large, naturally contiguous areas would introduce a degree of**

Table 1: Power-law coefficients for the four surface types, with "Connected segments" denoting the original segmentation and "Broken segments" for the finer SAM segmentation.

**arbitrariness. To address the reviewer's concern and to provide full transparency, we performed an additional analysis. We calculated the segment size distribution statistics using the finer-grained initial segmentation from the Segment-Anything Model (SAM) before merging segments of the same class (as shown in the manuscript's Figure 7c). The results, summarized in Tab. 1 of our response, show that the power-law coefficients change by 4-9% depending on the surface type. Further, in Fig. S4 we show a comparison between a) the "connected segmentation" and b) the "broken segmentation", which reveals a more distinct behaviour for all four surface types.**

| Surface type | Open Water | Ice-water mix | Thin ice | Snow-covered ice |
|---|---|---|---|---|
| Connected segments | -1.69 | -1.60 | -1.25 | -1.50 |
| Broken segments | -1.84 | -1.69 | -1.32 | -1.56 |
| Relative difference (%) | 8.9 | 5.6 | 5.6 | 4.0 |

**This confirms the FSD is sensitive to the segmentation approach, as the reviewer correctly pointed out. We have added this sensitivity analysis to Section 4.3 of the manuscript. Furthermore, to allow other researchers to explore alternative segmentation strategies, we have provided both the final, merged segmentation and the initial, finer SAM segmentation in our public dataset (Müller et al., 2025)**

[Figure]

Figure S4: Segment size distribution for a) the connected segments from the manuscript and b) the finer segmentation from SAM.

*Ln 227: We recognize that this merging step can connect large, contiguous areas of a single surface type (e.g., thin ice), which can influence feature size statistics. Therefore, the initial, finer-grained segmentation from SAM (prior to merging) is retained for a sensitivity analysis (see Sect. 4.3). To allow for full transparency and further exploration by the community, both the initial and final merged segmentation masks are provided in*
*our public dataset (Müller et al., 2025)*

*Ln 282: A key characteristic observed during our flights over the MIZ was the presence of large, contiguous areas of thin ice or snow-covered ice, which can appear as very large segments in our final classification (e.g., Fig. 7e). We interpret these as genuine physical features of newly forming ice in the MIZ at the time of observation rather than as*
*segmentation artifacts. However, we acknowledge that the definition of a "segment" is sensitive to the processing methodology and that the scale of these large features can influence the resulting feature size distribution (FSD). To quantify this sensitivity, we performed an additional analysis by calculating the FSD statistics on the initial, finer-grained segmentation generated by SAM, before our final step of merging adjacent*
*segments of the same class (an example of this initial stage is shown in Fig. 7c). The results, summarized in Tab 5, demonstrate that the power-law coefficients change by 4 to 8%, depending on the surface type.*

**RC1:**

I would recommend the authors to consider to address 2 points: (1) the potentials and possible challenges if one would extend this work to a wider time frame with stronger spatial variability which can have a bigger potential to impact the surface energy budget?

**Of course extending the method to a wider area is in general possible if the general characteristics of the surface types in this area are similar to our training data. A stronger variability of the same surface types does not affect the algorithm. However, adjustments, refinements and an extension of the training data will be needed when additional surface types, e.g., melt ponds, are present. Similar the**

**application to observations in other seasons will be challenging. E.g. in summer when skin temperatures of sea ice and sea water converge to -1.7°C a distinction from thermal IR observations becomes impossible. This would require the integration of data from other sensors (e.g., hyperspectral or microwave sensors) to improve classification accuracy. We addressed this points together with the**

**second comment in Section 5 of the manuscript.**

(2) the classification scheme is a great start and provides an inspiring direction for the arctic community, but i wonder if it is realistic to upscale this method to provide a pan-arctic insight?

**We thank the reviewer for this question. While direct upscaling of our airborne**

**measurement technique to a pan-Arctic scale is not feasible due to its limited spatial and temporal coverage, the methodology and findings offer a crucial pathway for improving pan-Arctic analyses. Our high-resolution data could serve as "ground truth" for calibrating and validating satellite-based products. As shown in our analysis, our dataset is increasingly dominated by snow-covered ice further**

**away from the sea-ice edge. This makes it difficult for a model trained solely on this data to learn the full heterogeneity of the marginal ice zone (MIZ), reinforcing the challenge posed by the limited spatial and temporal coverage of a single airborne campaign.**

*Ln 323: Extending this analysis to different seasons will be crucial for capturing*

*processes like melt pond evolution, though this will require multi-sensor data fusion to resolve the increased complexity of surface types. Therefore, the primary value of this high-resolution methodology lies in providing "ground truth" for calibrating satellite retrievals and refining sub-grid-scale parameterizations in pan-Arctic models.*